# Spatial Heterogeneity and Regional Clustering of Factors Influencing Chinese Adolescents’ Physical Fitness

**DOI:** 10.3390/ijerph20053836

**Published:** 2023-02-21

**Authors:** Zihan Tong, Zhenxing Kong, Xiao Jia, Jingjing Yu, Tingting Sun, Yimin Zhang

**Affiliations:** Key Laboratory of Exercise and Physical Fitness, Ministry of Education, Beijing Sport University, Beijing 100084, China

**Keywords:** physical fitness, influencing factors, spatial heterogeneity, regional clustering, K-means, multi-scale geographically weighted regression (MGWR), social–ecological model

## Abstract

There is often significant spatial heterogeneity in the factors influencing physical fitness in adolescents, yet less attention has been paid to this in established studies. Based on the 2018 Chinese National Student Physical Fitness Standard Test data, this study uses a multi-scale, geographically weighted regression (MGWR) model combined with a K-means clustering algorithm to construct a spatial regression model of the factors influencing adolescent physical fitness, and to investigate the degree of spatial variation in the physical fitness of Chinese adolescents from a socio-ecological perspective of health promotion. The following conclusions were drawn: the performance of the youth physical fitness regression model was significantly improved after taking spatial scale and heterogeneity into account. At the provincial scale, the non-farm output, average altitude, and precipitation of each region were strongly related to youth physical fitness, and each influencing factor generally showed a banded spatial heterogeneity pattern, which can be summarized into four types: N–S, E–W, NE–SW, and SE–NW. From the perspective of youth physical fitness, China can be divided into three regions of influence: the socio-economic-influenced region, mainly including the eastern region and some of the central provinces of China; the natural-environment-influenced region, which mainly includes the northwestern part of China and some provinces in the highland region; and the multi-factor joint-influenced region, which mainly includes the provinces in the central and northeastern regions of China. Finally, this study provides syndemic suggestions for physical fitness and health promotion for youths in each region.

## 1. Introduction and Literature Review

### 1.1. Current State of Physical Fitness

Physical fitness, as a measure of the ability to perform activities efficiently in daily life, is also an important component of promoting a healthy and active lifestyle [1,2]. Peak physical fitness in humans usually occurs during adolescence and continues [3,4]. Thus, the reduction in physical fitness that occurs during adolescence can pose a significant threat to the national public health system [5,6]. There are many reasons for the decline in physical fitness; socio-economic conditions, geography, and local government policies are considered as external factors that affect physical fitness [7]. In addition to this, overweight and obesity caused by high body fat percentage in individuals, and an undesirable lifestyle, including sedentary behavior, insufficient physical activity, and excessive video screen time, are intrinsic factors that may lead to lower physical fitness in children and adolescents [8,9]. Currently, there are approximately 1.2 billion adolescents worldwide, and their numbers are rising and are expected to peak around the 2030s, when they will account for one-sixth of the world’s total population [10]. However, over the past few decades, the physical fitness of children and adolescents worldwide has not been encouraging. A review of a sample size of more than 100,000 children and adolescents indicated that global adolescent physical fitness declined by nearly 10% between 1975 and 2000, with particularly significant declines in endurance and flexibility qualities [11,12]. The same downward trend in exercise performance (both anaerobic and aerobic capacity) has also been found by researchers [13]. After entering the 21st century, most studies from various countries show that the overall situation is still very serious, although the declining trend of children and adolescents’ physical fitness has been reduced [14]. The decline in adolescent physical fitness has been particularly pronounced in Asia [15], where the total increase in body fat among children from the 1950s to the end of the 20th century was more than twice that of other countries. Even after the 21st century, adolescent body fat levels have stabilized in some countries, including Europe and America [16,17]; however, in most Asian countries, adolescent body fat levels continue to increase, which indirectly leads to a further decline in the physical activity and physical fitness levels of these adolescents [18,19]. The huge size of the population and its potentially important influence have led to an increased focus on the assessment of adolescent physical fitness and its interventions in recent years in various countries as well. The whole life-cycle theory provides an important perspective for relevant public health actions, and health interventions during adolescence can provide a solid foundation for future physical fitness and health levels in adulthood [20]. It is worth noting that adolescent fitness is not only influenced by genetic factors, but also closely related to the living environment in which the individual lives, such as the adolescent’s family context, the community in which the youth lives, the school, and the geographic location at the macro-spatial scale [21,22]. All these elements can influence the planning and implementation of health interventions. Therefore, clarifying and establishing quantitative models of the relationship between multiple influencing factors and adolescent physical fitness is particularly important for the development of intervention strategies that effectively promote adolescent physical fitness development [23].

### 1.2. Factors Influencing Physical Fitness

The ecological model of health promotion is considered an effective research method for sustainability in the field of public health [24]. The theoretical framework and adaptive model underlying this approach are flexible and modifiable. The ecological model of health promotion allows for a more integrated and comprehensive approach to physical fitness and health issues than health benefit promotion approaches that focus on a single dimension [25]. Environmental factors that influence human health have multi-dimensional properties, and the environment can be classified into different dimensions according to specific research objectives, principles, and criteria. For example, according to the composition of environmental elements, these can be divided into atmospheric environment, soil environment, marine environment, and other types; according to the different spatial scales, the environment can be divided into global environment, regional environment, local environment, etc.; according to the dominant way of human use, the environment can be divided into urban environment, rural environment, and socio-economic environment, etc. [26,27]. WHO attributes up to 24% of global deaths to the environment [28]. Assessment reports from around the world also show different trends in the physical fitness of youths in different geographic locations [29,30]. The effects of geographic location include the natural and built environment in which humans live and the socio-economic environment that has developed based on it. In recent years, the number of empirical studies on the effects of environmental changes on physical fitness due to different geographical locations has started to increase. Overall, previous research has focused on individual behavior, family environment, and school [31,32,33]. Most of the effects caused by these small-scale environmental factors are related to the culture of the country or region where they are located and the educational background, and are part of the social environment. As research on adolescent physical fitness has intensified, social factors such as economics, migration, and urbanization have received progressively more attention. Existing studies have shown that socio-economic factors can influence the physical fitness and health of a population to some extent [34,35,36]. Children and adolescents living in higher socio-economic development and without immigrant backgrounds have higher physical fitness [37,38]. The effect of urbanization on adolescent physical fitness and health remains controversial. On the one hand, accelerated urbanization brings more healthcare opportunities and sports facilities to the population, which are beneficial for physical fitness and health promotion, but on the other hand, rapid urbanization changes the lifestyle of the population, such as prolonging sedentary and other undesirable behaviors. Extreme weather and environmental pollution associated with urbanization has also reduced the amount of time people spend outdoors, which has a negative impact on physical fitness [39,40]. Population health is inextricably linked to climate change and the natural environment [41,42,43]. Some of the scholars’ studies emphasize the mechanisms of the natural and built environment that influence adolescents’ physical fitness. For example, the altitude of the area where they are located is the main natural environmental factor influencing the development of physical fitness and form in children and adolescents [44,45,46]. Existing studies have shown that there is a highly significant correlation between human aerobic capacity and altitude, and that appropriate altitude can effectively enhance human aerobic capacity [47]. At the same time, children living at high altitudes have higher levels of physical activity relative to children living at lower altitudes because the expansion of the built environment at lower altitudes imposes many spatial constraints on children’s activities [46,48]. The area of green space coverage is not only related to physical fitness and health behaviors, but is also considered to be one of the determinants of the prevention of mental illness [43,49]. At the same time, studies on the built environment and adolescent physical fitness have shown that the intensity of road traffic is significantly and negatively associated with adolescent physical fitness, while positive changes in the built environment can contribute to the improvement of adolescent physical fitness through the mediating effect of physical activity [50,51,52]. From the perspective of the social ecology of health promotion, since individual behaviors always occur in the environment in which they are located and there are interactions between individuals and the environment, it is necessary to give attention to the complexity and dynamic interactions of different categories of environmental factors and to advocate a multi-scale, interdisciplinary, and syndemic approach to research [53].

### 1.3. Hypotheses and Objectives

Environmental factors affecting adolescent physical fitness are often geographically and spatially differentiated, and research on the spatial heterogeneity of adolescent physical fitness drivers is imperative. On the one hand, adolescents in different geographic regions have different physical forms, physiological functions, and motor qualities, and although these indicators together constitute physical fitness, they are differentiated into different weights due to changes in location [54,55]. On the other hand, the factors influencing physical fitness are not completely mobile in space. As the location of geospatial regions changes, the influence of various environmental factors on body mass also change, with a non-global character [56]. It is due to the existence of spatial heterogeneity that if a uniform standard method is used to assess physical fitness indicators for various populations that are in different geographic settings, the impact caused by locational differences between various types of areas may be overlooked, resulting in an inability to assess the physical fitness status of adolescents correctly and effectively [57]. Nevertheless, due to the limited data acquisition, there is still a lack of research on the spatial heterogeneity of factors influencing adolescent physical fitness at macro-spatial scales. In addition, the identification of regions based on the spatial heterogeneity of influencing factors and thus the construction of a comprehensive intervention applicable to adolescent physical fitness is also an urgent task [58,59]. For possible spatial heterogeneity, the traditionally used global regression models (e.g., OLS models) lack explanations for spatial variability and ignore causal relationships between independent and dependent variables in local geographic areas, whereas geographically weighted regression (GWR) accommodates the existence of spatial heterogeneity [60].

Based on this, this study used Chinese regional characteristic indicators with the physical fitness of the study population—junior secondary school students (12–15 years old)—in selected regions to investigate the spatial heterogeneity of factors influencing adolescent physical fitness across regions at the provincial scale in China. This was performed by using a multi-scale, geographically weighted regression model (MGWR) and applying a K-means clustering algorithm for clusters of regions with similar influencing factors. This work is expected to quantify the disparity in youth physical fitness development in different geographical locations and provide a reference for future thematic area identification in public health.

## 2. Methodology

### 2.1. Multi-Scale, Geographically Weighted Regression

Because the model parameters constructed by ordinary least squares (OLS) are global, they do not reflect well the spatial heterogeneity of parameters that exist over different geographic spaces. The variability that exists in spatial data should be taken into account in conducting the analysis of youth physical fitness problems in different regions [61]. This study introduces a spatial regression model to investigate the possible spatial heterogeneity in the influencing factors of adolescent physical fitness. There are several types of spatial regression models, and the classical geographically weighted regression model (GWR) is based on a fixed bandwidth for regression analysis, which assumes the same scale of spatial heterogeneity for each type of influencing factor and the same range of influence at different spatial locations. The results of GWR may lead to bias in the final parameters, making them less explanatory for the spatial heterogeneity of influencing factors of student fitness in different regions [62]. Multi-scale, geographically weighted regression (MGWR), on the other hand, relaxes the assumption of equal spatial scales, allowing the bandwidth in the model to adjust according to different explanatory variables [61,63]. Therefore, MGWR can effectively optimize and avoid analytical errors caused by the shortcomings of previous geographically weighted regression models due to the influence of scale homogenization [64]. This study first obtained the correct explanatory variables by constructing OLS, after which MGWR was used to explore the spatial relationship between student physical fitness and explanatory variables. The MGWR equation is shown below:(1)$$y_{i}=\overset{m}{\underset{j=0}{\sum }}\beta_{bw_{j}}(u_{i},v_{i})x_{ij}+\varepsilon_{i}$$

In Equation (1), $y_{i}$ is the physical fitness of students in region *j*, $\beta_{bw_{j}}$ is the most appropriate sample bandwidth in region *j*, $x_{ij}$ is the observation of the j variable at i, and $\varepsilon_{i}$ is the random error term. The kernel function and bandwidth selection criteria for MGWR refer to the GWR determination criteria, and the bandwidth is determined using the corrected Akaike information criteria (*AICc*). The *AICc* is a measure of model performance that is useful for comparing different regression models. Given the complexity of the model, a model with a lower *AICc* value will fit the observed data better and is useful for comparing models that apply to the same dependent variable and have different explanatory variables [65].
(2)$$AIC_{C}=2n\ln\left(\sigma \right)+n\ln\left(2\pi \right)+n\frac{n+\mathrm{tr}\left(S\right)}{n-2-\mathrm{tr}\left(S\right)}$$
where n represents the number of sample observations, σ represents the standard deviation of the error term, and tr(S), as a bandwidth function, is expressed as the trace value of the S matrix in the regression model.

### 2.2. K-Means Clustering

Clustering analysis is a technique for statistical data analysis. Cluster analysis has received wide application in many fields, including machine learning, data mining, pattern recognition, image analysis, etc. [66]. Clustering is the division of similar objects into different groups or more subsets by static classification. Spatial clustering is an important part of spatial data mining, which mainly clusters or classifies entities according to their characteristics; identifies clusters or densely distributed regions in large-scale multi-dimensional spatial data sets according to a certain distance or similarity measure; and divides the data into a series of mutually distinguishable groups to find the overall spatial distribution patterns and typical patterns of the data sets [67]. This study was based on the Calinski–Harabasz pseudo-F index to measure the within-group similarity and between-group variability of different grouping approaches and to spatially cluster the regional characteristic indicators affecting adolescent physical fitness. The K-means algorithm uses the mean of all data samples within each cluster subset as a representative point for that cluster, and its main idea is to optimize the criterion function for evaluating the clustering performance by dividing the data set into different classes through an iterative process. Each cluster generated by the algorithm has similar intra-group attributes and independent inter-group attributes [68]. If the Calinski–Harabasz pseudo-F index is high, it means that the intra-group distance of the clustering results is small while the group spacing is large, and the clustering results are more reliable. The equation is shown below:(3)$$F=\frac{\frac{R^{2}}{n_{c}-1}}{\frac{1-R^{2}}{n-n_{c}}},\;\;R^{2}=\left(SST-SSE\right)/SST$$

In the equation, *n* denotes the number of elements and $n_{c}$ denotes the number of classes (groups). SST reflects the difference between groups. SSE is the statistic reflecting the similarity within groups. In this study, the K-means clustering algorithm was used to identify areas of similarity in the relevant factors that have an impact on adolescent physical fitness, and for each clustered area of influence, a syndemic public health guideline can be proposed.

## 3. Data Collection and Analysis Process

### 3.1. Data Sources

The physical fitness data were obtained from the 2018 Chinese National Student Physical Fitness Standard (CNSPFS) survey conducted by the Ministry of Education of the People’s Republic of China. Physical fitness data were obtained according to the randomization principle, and combined stratified and whole-group sampling was used to draw equal proportions of samples from 31 provinces (excluding Hong Kong, Macao, and Taiwan) in mainland China; the specific sampling schemes can be found in previous literature [69]. The final number of valid samples obtained from the screening of this study was 48,922. According to the evaluation criteria of CNSPFS, students’ physical fitness status are classified into three grades according to the composite score of physical fitness test: high grade (≥80), medium grade (≥60 and <79.9), and low grade (<60). The dependent variables in this study were the detection rates of the three grades of high, medium, and low physical fitness of middle school students in 31 provinces in mainland China in 2018, respectively, and the detection rate of physical fitness grades in each region was equal to the percentage of the number of corresponding grades in that region to the total sample size in that region.

The physical fitness test protocol for students in mainland China mainly includes three aspects: physical form, physiological function, and motor quality. The test indexes in this study included seven items: BMI, lung capacity, 50-m sprint, seated forward bend, standing long jump, pull-ups/sit-ups, and endurance running. The scores of each individual index were calculated for each student with reference to the gender and age scoring criteria in CNSPFS. The students with excellent performance in pull-ups/sit-ups and endurance running were given extra points, which were determined according to the actual test results, with an upper limit of 20 points. Finally, the overall score of each student was calculated according to the weight of individual indexes. Physical Fitness Score = BMI × 0.15 + lung capacity × 0.15 + 50 m run × 0.2 + seated forward bend × 0.1 + standing long jump × 0.1 + pull-ups/sit-ups × 0.1 + endurance run × 0.2 [70,71]. Prior to testing, all potential participants provided written informed consent from the school and parents to ensure that the participating student was free of disease and injury and fit to take the physical fitness test. All testing instruments were consistent at each testing site and were calibrated prior to use. All staff members are pre-trained, and they have passed relevant tests. School nurses and third-party monitors are available at the testing sites to prevent injuries to subjects during the testing and to monitor that the physical fitness tests are being conducted properly according to CNSPFS standards.

This study attempts to explain the degree of variability in adolescent physical fitness from a socio-ecological perspective. The explanatory variables were selected mainly around two dimensions: geographical environmental factors (including natural environmental variables and built environment variables) and socio-economic factors that can represent the regional development status (including regional economic variables, urban attribute variables, cultural and educational variables, healthcare variables, and sports development variables). Finally, a total of 12 regional characteristic indicators that may affect the regional adolescent physical fitness status were obtained (Table 1). The theoretical basis for the selection of explanatory variables is mainly derived from the socio-ecological model of health promotion, that is, the health status of a population is always influenced by the combination of the ecological environment, economic environment, and social policies in which it is located [72,73]. The indicators of environmental and socio-economic indicators together with other regional characteristics selected for this study were obtained from the official data released by the National Bureau of Statistics of China.

### 3.2. Model Construction

An OLS model of youth physical fitness status and regional characteristics indicators was first constructed in SPSS 26.0 to obtain the correct explanatory variables and to verify the correlation between potential influences such as environment and socio-economic factors of each region and youth physical fitness. The OLS model needed to satisfy the conditions that the data had chi-square and normal distribution, and that there was no multi-collinearity among the independent variables [74]. In this study, variance inflation factor (VIF) <5 and Shapiro–Wilk test of significance *p* > 0.05 were selected as screening conditions to further extract the eligible explanatory variables (Table 2). There is an association between some regional characteristic indicators and the physical fitness of junior high school students. The preliminary normality test of these elementary indicators showed that variables such as percentage of forest area, non-farm output, and road density satisfied normal distribution alone and could be directly included in the regression model for calculation. Some of the explanatory variables that do not conform to the normal distribution, such as the average altitude of the provinces, the annual precipitation, and the area of school playgrounds, cannot be directly included in the regression model for calculation. Therefore, in this study, these explanatory variables were made to better conform to the normal distribution by taking the natural logarithm [72].

In this study, the explanatory and dependent variables after pre-processing were included and three OLS models were constructed (Table 3). From model 1, it was found that the physical fitness high-grade rate (EGR) was significantly and positively influenced by the annual precipitation (Rain) and non-farm structure ratio of the region (non-farm output). Junior school students living in areas with relatively higher precipitation, wetter natural environment, and more developed secondary and tertiary economic industries had better physical fitness. As shown in Model 2, the physical fitness mid-grade rate (PR) was significantly and positively correlated with the average altitude of the region, the PR was deeply influenced by the natural environment, and the junior high school students with a composite score in the range of 60–79.9 on the physical fitness test were generally distributed in regions with relatively high average altitude. From model 3, the physical fitness low-grade rate (FR) is significantly and negatively correlated with the average altitude of the region (Altitude), the annual precipitation (Rain), and the ratio of non-farm structures in the region (non-farm output). If adolescents live for a long time in an area with low precipitation, low altitude, and where agriculture is the main source of economy, it may be detrimental to their physical fitness improvement and development.

To determine whether the explanatory variables exhibited spatial heterogeneity in the regression model, after obtaining appropriate explanatory variables for adolescent physical fitness status through OLS models, the same explanatory variables were incorporated into MGWR2.2 to construct a multi-scale, geographically weighted regression model of adolescent physical fitness and influencing factors using the detection rates of three physical fitness classes, high, medium, and low, as the basis for classification (Table 4). Comparing the results of the MGWR and OLS models, the R^2^ of the adolescent physical fitness regression model improved significantly after considering the spatial local characteristics and the changes in spatial scales. The AICc and the residual sum of squares of each model also show a decreasing trend, indicating that the MGWR model has better ability to fit and explain the spatial pattern of variables, and is therefore better than the OLS model. As a local regression model, the variability variables in MGWR have unique R^2^, standard errors, and *t*-values at each sample point. In summary, the reliability of the MGWR model was further validated by the comparison results from the regression analysis model. Through the MGWR model, this study further analyzes the degree of differentiation across sample areas and the spatial pattern of drivers influencing youth physical fitness status. To explain the spatial divergence patterns of the influencing factors more effectively, the impact coefficients of the explanatory variables in the MGWR model of EGR, PR, and FR were selected to satisfy the significance test (|t| ≥ 1.96) for statistical analysis in this study. Spatially divergent pattern maps of the influencing factors were produced in ArcGIS 10.7, and the degree of influence for each variable was classified into five categories based on color according to Jenks’ method: highest, relatively higher, medium, relatively lower, and lowest.

## 4. Results and Discussion

### 4.1. Spatial Patterns of Influence Factors

The results showed that the factors influencing the physical fitness of adolescents showed a significant pattern of banded spatial heterogeneity (Figure 1). The non-farm industrial structure ratio, also known as the non-farm output ratio, is calculated as the ratio of the production value of secondary and tertiary industries to the gross domestic product of the region. When upgrading of regional industrial structures accelerates, the ratio of non-farm industrial structures also increases. The non-farm industrial structure ratio is also the main explanatory variable for evaluating the social development and economic changes in a region [75]. In this study, firstly, from the MGWR model of EGR (Figure 1a), the effect of non-farm output on the physical fitness high-grade rate of students is positive, taking values between 0.539–0.544, with a mean value of 0.542. This indicates that for every 1 percentage point increase in the non-agricultural structure ratio, the improvement space of the physical fitness high-grade rate of youth in the region is between 0.539–0.544 percentage points, with different increases in different regions and an average increase of about 0.542%. From the spatial pattern distribution for the coefficients, the influence of non-farm output on EGR shows an E–W directional spatial pattern in general, with the variable coefficients increasing in a gradient from west to east. The highest influence is found in Fujian and Zhejiang provinces in the eastern coastal region, and in the northeastern region (Heilongjiang, Jilin, and Liaoning). In western China, such as Xinjiang and Tibet, the non-farm output has a weaker contribution to EGR, which reflects the variability in the positive influence mechanism of the non-agricultural industrial structure ratio on the youth high physical fitness population in different regions. In the PR model (Figure 1c), the non-farm output had a statistically significant effect on the physical fitness mid-grade rate. The values ranged from −0.419 to −0.207, with a mean value of −0.339. From a statistical point of view, the non-farm output had an overall negative effect on the physical fitness mid-grade rate of the district youth; however, the decrease in the rate of medium physical fitness scores does not necessarily mean a decrease in the overall level of student fitness in the region, and should be evaluated specifically in relation to the overall change in the high-grade rate as well as the low-grade rate of student physical fitness in the region. From the spatial pattern distribution for PR model coefficients, the variable coefficients show an N–S spatial pattern. It is mainly the provinces in southern and some central regions of China whose coefficients of non-farm output are statistically significant, and their negative influence gradually decreases from north to south. From the MGWR model of FR (Figure 1e), the effect of non-farm output on the low-grade rate in youth physical fitness in each region is negative, taking values between −0.841 and −0.394, with a mean value of −0.530. In terms of coefficient absolute value, the range of variation in the intensity of its effect is the largest among all variables. The coefficient changes drastically with the change in spatial location. On average, for every 1% increase in the rate of non-farm output, the low-grade rate of student physical fitness in the area decreases by 0.53%. The coefficient shows a NE–SW spatial pattern in China’s significant provinces, with the negative influence of the non-farm output gradually decreasing from the northeast toward the southwest. It is worth noting that the highest negative influence of non-farm output is still in the northeast (Heilongjiang, Jilin, and Liaoning), which indicates an urgent need for industrial structure upgrading in this area, and, on the other hand, it also explains the geographic reasons for low physical fitness and more serious overweight and obesity in youths in the Northeast China for a long time.

Precipitation is the depth of liquid or solid (after melting) water that falls from the sky to the ground during a certain period and accumulates on the water surface without evaporation, infiltration, or loss. In meteorology, the amount of rain and snow that falls in a year and is collectively melted into water is called annual precipitation. Precipitation is an important factor in measuring climate change. This study found that annual regional precipitation has an important effect on physical fitness among many natural environmental factors. Precipitation was statistically significant in both the EGR and FR models, with a positive effect on the physical fitness of adolescents in each region. In the EGR model (Figure 1b), the coefficient of influence varies between 0.448 and 0.554, with a mean value of 0.503. This indicates that for every 1 mm increase in annual precipitation in each region of China, the average increase in the high-grade rate of physical fitness of junior high school students in that region is close to 0.503%. From the MGWR model of FR (Figure 1g), the range of variation in the influence of precipitation coefficient is between −0.726 and −0.719, with a mean value of −0.724. This indicates that the annual precipitation has a large influence on the failure rate of student physical fitness, with for each 1 mm decrease, the average increase in the rate of low physical fitness in the region is close to 0.724%. In terms of the regional distribution in annual precipitation, the coastal areas of southeast China are the first to receive water vapor from the southeast monsoon, resulting in abundant precipitation. The northwestern region is deep inland and far from the sea, so the annual precipitation in China generally shows a decreasing pattern from southeast to northwest. The spatial pattern of coefficient influence also corresponds to the same pattern. In both the EGR and FR models, the influence of precipitation on body mass shows a decreasing trend from southeast to northwest, and the areas with the greatest change in physical fitness are mainly concentrated in the area southeast of the 800 mm iso-rainfall line.

Changes in the natural environment brought about by altitude can affect the human body in a variety of ways. The most important reaction to the arrival from the plain at a lower altitude to the plateau zone at a higher altitude is the hyperventilation of the lungs caused by the decrease in the partial pressure of oxygen, and long-term exposure to the plateau area causes changes in the morphological development of the body [76]. In a certain altitude range, the body can tolerate hypoxia through various adaptations to the low oxygen environment, and the individual’s physical fitness, especially physiological functions, will change. The changes mainly include physiological as well as metabolic adaptations, increased pulmonary ventilation and regulation of acid–base balance in the body, increased erythropoiesis and changes in local circulation and cellular metabolism, which clearly favor oxygen transport and utilization [77]. According to the results obtained from this study, altitude was a significant influence factor in the PR model and in the FR model. For Chinese adolescents’ mid-grade physical fitness rate, the influence of altitude ranged from 0.638 to 0.642, with a mean value of 0.641. For every 1 m increase in altitude, the mid-grade physical fitness rate of regional adolescents increased by 0.641 percentage points. The altitude of each province in China basically shows a gradual decreasing trend from northwest to southeast; however, the spatial pattern of altitude coefficients from the PR model (Figure 1d) shows that the influence of altitude on youth physical fitness gradually increases from northwest to southeast direction, which is opposite to the trend of altitude in China. This phenomenon may be because the altitude in the northwestern part of China is already at a high plateau and sub-plateau, and even though there are differences in altitude in this part, the overall altitude is at a high level, thus causing a diminishing marginal effect on the change in physical fitness. In the southeast of China, especially along the coast, the altitude is generally lower and in the plains. Even small changes in altitude can lead to large changes in coefficient effects between provinces in this topography. From the FR model (Figure 1f), the effect of mean altitude ranged from −0.554 to −0.551 with a mean value of −0.552, which had a negative effect on the rate of low physical fitness rating. It shows that the rate of low physical fitness in each province decreased by 0.552% on average with an increase of 1 m in altitude. The spatial pattern of the influencing factors shows that the negative influence of altitude gradually decreases from northwest to southeast of China. The reason for this phenomenon may be due to the fact that the rate of low physical fitness grade is affected by a combination of influencing factors, and currently still depends mainly on the socio-economic development conditions. For the northwestern region of China, owing to the relatively weak level of economic and social development, the provinces’ physical fitness status is mainly influenced by the natural environment. In contrast, owing to the high level of urbanization and the relatively better economic conditions in the eastern coastal regions, the influence of low fitness rate by altitude is weakened.

### 4.2. Identification and Classification of the Influence Areas

Through the construction of OLS and MGWR models, key factors influencing the physical fitness of junior high school students in each region of China were uncovered. Based on this, this study used the K-means multivariate clustering algorithm to cluster each variable of variability to identify the areas of influence on youth physical fitness and to cluster each provincial area in China. The validity of the final clusters was measured by the Calinski–Harabasz pseudo-F index, and the value of the statistic was calculated separately for each clustering scheme (Figure 2). The value of the pseudo-F index was maximum when k = 3 (Figure 3). Therefore, clustering analysis was performed when k = 3 was set.

The results of multiple clustering show that provinces in the same category exhibit obvious spatial clustering characteristics (Figure 4). According to the differences in the factors influencing the physical fitness status of adolescents in different types of zones (Table 5), region I contains 11 provinces and cities: Beijing, Tianjin, Shandong, Henan, Chongqing, Hubei, Anhui, Jiangsu, Shanghai, Zhejiang, and Guangdong. Compared to other regions, this region has the lowest average altitude, the highest percentage of non-farm output, and the highest road network density. Most of the geographical locations in region I are in the plains (mainly between the central and eastern parts of China) and eastern coastal areas, and generally have a high degree of socio-economic development, more urbanized development, and convenient transportation, so region I can be regarded as a socio-economic-influenced region. Region II contains seven provinces: Inner Mongolia, Shanxi, Ningxia, Gansu, Qinghai, Xinjiang, and Tibet. For region II, it can be clearly found that it is mainly influenced by the natural environment, the mean value of altitude is the highest, the area of forest cover is also significantly lower than in other regions, and the socio-economic development is lower; most of the provinces are mainly within the western region, the geographical characteristics are more obvious, and it can be seen as a natural-environment-influenced region. Region III contains 13 provinces: Heilongjiang, Jilin, Liaoning, Hebei, Shaanxi, Sichuan, Yunnan, Guizhou, Hunan, Jiangxi, Fujian, Guangxi, and Hainan. The mean values of each influencing factor of youth physical fitness in this region basically did not contain maximum or minimum values. Compared with the other two types of regions, the development of influencing factors in region III is more balanced, and most of the provinces included belong to central China and some northeastern regions, which can be regarded as a multi-factor joint-influenced region. The K-means algorithm enables multivariate clustering of factors influencing adolescent physical fitness, which can identify areas affected by similar factors, and thus provides a basis for regional scientific physical fitness promotion guidance programs and the realization of health-oriented urban planning policies.

## 5. Conclusions

This study uses a multi-scale, geographically weighted regression model, supported by the 2018 China Student Physical Fitness Standards (CNSPFS) sample review data, for the spatially divergent patterns of factors influencing the physical fitness status of adolescents at the provincial scale in China, and finally a regional clustering exercise using the K-means method for each region of China based on the model results, leading to the following main conclusions:

1. From the model results, the multi-scale, geographically weighted regression model (MGWR) results are more reliable than the ordinary least squares model (OLS). The results of the OLS model in this study did not identify the statistical significance of the non-farm output in the rank rate for physical fitness, but the MGWR found a significant influence of this factor in southern China, so considering the spatial heterogeneity and spatial scale of the influence factor affects the model results and analysis of the youth physical fitness study.

2. Among the three multi-scale, geographically weighted regression models constructed, for the model of the physical fitness high-grade rate, the strength of influence for each indicator from the largest to the smallest is the non-farm output and the annual precipitation of each region, the regression coefficients are all positive. Among them, the spatial heterogeneity of annual precipitation is strong, so that the effect of this factor on the physical fitness high-grade rate changes dramatically depending on the spatial location. In the model of the physical fitness mid-grade rate, the non-farm output has a high influence in local areas, while the average altitude has a higher influence on the physical fitness mid-grade rate in all regions of the country. However, the judgment of the regional physical fitness status cannot be made solely from the change in the number of people with intermediate fitness test grades. It should be analyzed together with the changes in the number of people who reach high and low fitness grades in the region. There are three statistically significant influencing factors in the model of low physical fitness rate, and all of them are negative for the model, among which the coefficient of influence of the natural environment has the largest absolute value, indicating that the natural environment of residence has a greater influence on the low physical fitness rate of Chinese adolescents.

3. According to the characteristics of the factors influencing youth physical fitness at the provincial spatial scale, the 31 provinces in mainland China can be clustered into three major influenced regions. Among them, Region I mainly includes provinces in eastern and some central regions of China, which are mainly influenced by social and economic factors. Region II mainly includes provinces in northwestern China and some highland areas, which are mainly influenced by natural environmental factors. Region III mainly includes provinces in the central and northeastern regions of China, and this region is influenced by a combination of factors.

### Planning Implications

In response to the above findings, the following suggestions can be made for the promotion and development of youth physical fitness in different regions: In terms of the development of youth physical fitness in China, the most important socio-economic influencing factor is the ratio of non-farm output. Combined with the spatial pattern of the non-farm output in mainland China, the most profound influence is in the eastern coastal region and the northeastern region of China. Therefore, to improve the physical fitness and health level of regional youths and reduce the social impact caused by the sub-standard physical fitness of regional youths, we should accelerate the upgrading of industrial structure in the region and promote the development of the national economy and urbanization. At the same time, the construction of the fitness environment should be strengthened to promote the positive interaction between social drivers, environmental drivers, and individual youth lifestyles. From the perspective of this study, the future improvement of regional students’ physical fitness can start from the idea of accelerating industrial restructuring to stimulate regional economic and social development and cultivate a healthy environment adapted to the development of adolescent physical fitness. The eastern region of China should maintain an open economy, make full use of the resource conditions of domestic and international markets, and continue to upgrade and transform its economic structure to maintain the momentum of the high regional physical fitness rate and drive the healthy development of adolescent physical fitness in the surrounding areas. The factors influencing youth physical fitness in Northeast China are more complex and should be given high attention in the future. A quantitative spatial model for the relationship between climate change, socio-economic development characteristics, and youth physical fitness in northeastern China can be established by narrowing the scope of the study and further refining the spatial scale to the county and city level.

In addition, it is necessary to face the fact that there are large regional differences in the physical fitness status of Chinese students and the factors influencing it, and to develop targeted public health guidance programs according to local conditions. It is necessary to establish sub-regional urban planning mechanisms and strengthen inter-regional linkages to fully consider the unevenness of various indicators of physical fitness due to different geographical locations. For region I, it is necessary to provide an advantageous fitness environment and conditions for young students living in this area, promote the upgrading and transformation of the industrial structure, and focus on strengthening the transformation of the built-up areas of the city, enhancing the construction of sports facilities and public health services, and approaching the development concept of a healthy community and even a healthy city. For region II, it is necessary to give attention to the impact of natural environmental factors on the physical fitness of young students, to implement more flexible scientific fitness guidance according to the changes in their natural environment, and to give more play to the geographical environment and climatic advantages of the region, while the construction of infrastructure should also receive attention. For region III, the influence of multiple factors on the physical fitness of junior high school students should be considered comprehensively to promote the rational allocation of resources from all parties. At the same time, for the multiplicity and complexity of factors influencing this type of area, future research should be increased on the influence mechanisms of youth physical fitness and health risks due to changes in the comprehensive regional characteristics of this type of area.

There are, however, still limitations to this study. Owing to the complex composition of physical fitness itself and the fact that the mechanisms influencing adolescent physical fitness are complex, various factors influence this issue in different ways. Therefore, the results of this study based on cross-sectional data and the MGWR model need to be fully validated by subsequent studies to determine whether there is a spatial and temporal lag effect of various influencing factors on adolescent physical fitness and whether changes in the spatial scale of the study area may interfere with the results to some extent.

## Figures and Tables

**Figure 1 ijerph-20-03836-f001:**
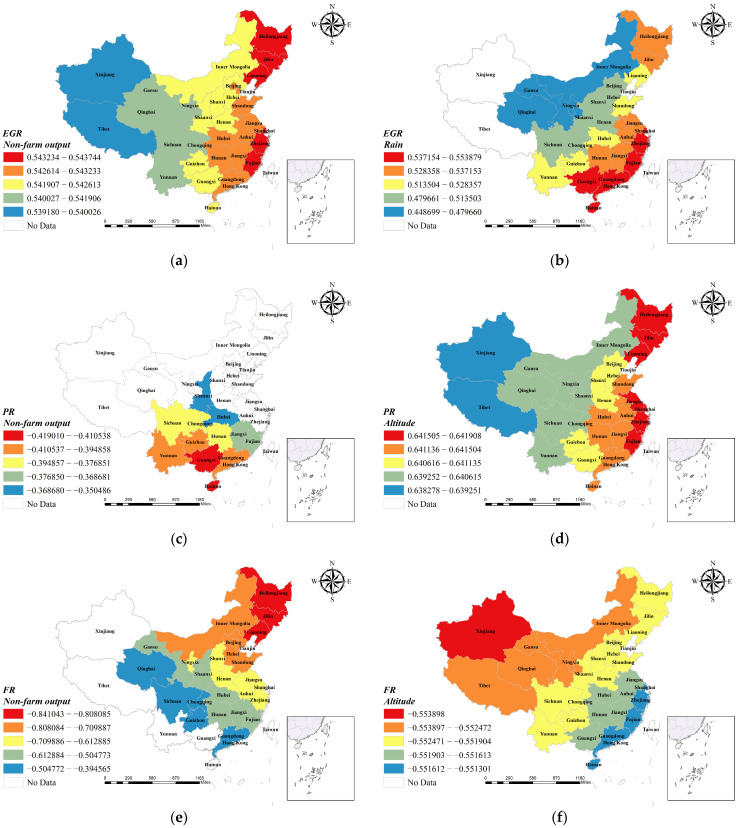
Spatial patterns of factors influencing Chinese adolescents’ physical fitness in 2018. (**a**) Non-farm output in the EGR group; (**b**) Rain in the EGR group; (**c**) non-farm output in the PR group; (**d**) Altitude in the PR group; (**e**) non-farm output in the FR group; (**f**) Altitude in the FR group; (**g**) Rain in the FR group.

**Figure 2 ijerph-20-03836-f002:**
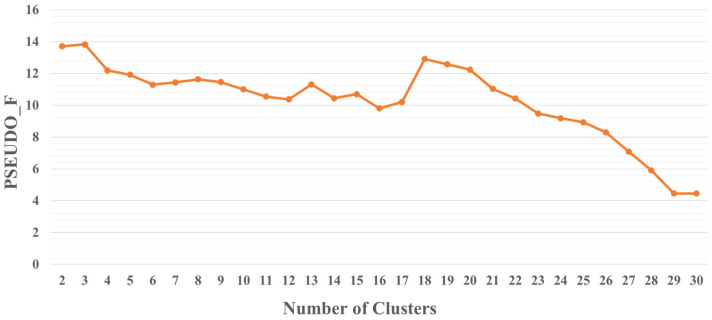
Pseudo-F index based region clustering basis.

**Figure 3 ijerph-20-03836-f003:**
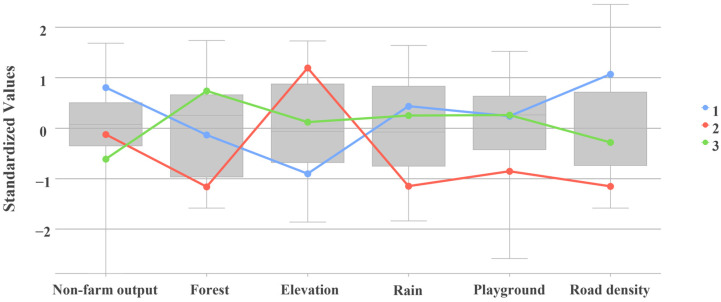
Standardized multivariate clustering boxplots.

**Figure 4 ijerph-20-03836-f004:**
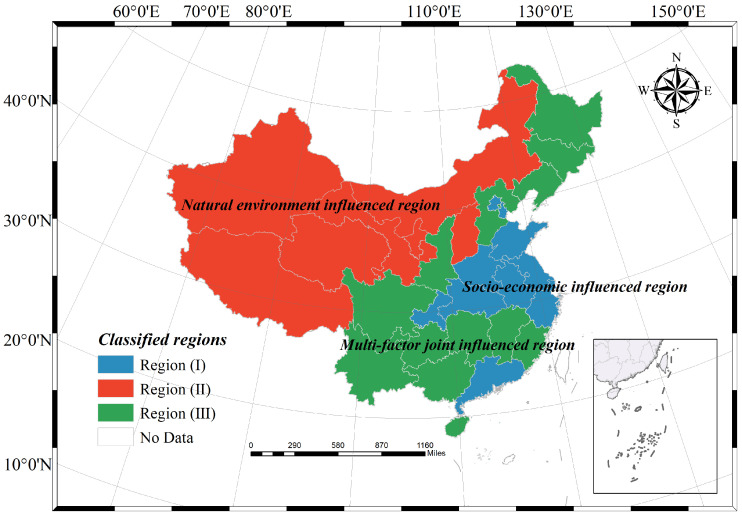
Regional clustering of factors influencing Chinese adolescents’ physical fitness.

**Table 1 ijerph-20-03836-t001:** Description of the variables and data sources.

Name	Description	Data Source
Dependent variables		
EGR	Percentage of youth with a physical fitness score ≥80 per province (%)	CNSPFS
PR	Percentage of youth with a physical fitness score ≥60 and <79.9 per province (%)	
FR	Percentage of youth with a physical fitness score <60 per province (%)	
Explanatory variables		
Altitude	Average altitude per province (m)	Chinese altitude barometer
Rain	Annual total precipitation per province (mm)	China Environment Statistics Yearbook 2018
Forest	Percentage of forest area per province (%)	
Temperature	Average air temperature (0.1 °C)	
Green space	Area of green space in parks per province (km^3^)	China Statistical Yearbook 2018
Playground	Sum of playground area of junior high schools (m^2^)	
Healthcare	Health care spending per province (yuan)	China Social Statistics Yearbook 2018
Non-farm output	The ratio of provincial non-farm output to GDP (%)	National Bureau of Statistics
Road density	Length of primary and secondary roads per sq. km. (%)	
Urban	The ratio of urban population to total population (%)	China Statistical Yearbook 2018
Education	Average number of years of education (year)	Bulletin of the National Population Census 2020
Athletes	Number of athletes above Level 2 per province	China Education Statistics Yearbook 2018

EGR = excellent and good rates of physical fitness; PR = passing rate of physical fitness; FR = failure rate of physical fitness.

**Table 2 ijerph-20-03836-t002:** Tests for normality and covariance of factors influencing physical fitness.

Variable Name	Shapiro–Wilk	*p*	Tolerance	VIF
Frost	0.937	0.070	–0.110	–0.109
Non–farm output	0.940	0.081	0.543	0.544
Road density	0.976	0.698	0.015	0.016
Altitude ^1^	0.967	0.438	–0.072	–0.069
Rain ^2^	0.935	0.062	0.525	0.554
Playground ^3^	0.944	0.106	0.145	0.149

^1,2,3^ Variables converted to natural logarithms.

**Table 3 ijerph-20-03836-t003:** Overall results of the three types of regression models (OLS).

OLS	Model 1: EGR		Model 2: PR		Model 3: FR	
	β	|*t*|	β	|*t*|	β	|*t*|
Frost	−0.127	0.696	0.047	0.258	0.159	0.821
Non-farm output	0.528	3.069	−0.279	1.624	−0.574	3.145
Road density	−0.019	0.083	0.102	0.458	−0.075	0.318
Altitude	−0.106	0.535	0.667	3.377	−0.519	2.471
Rain	0.470	2.250	−0.005	0.022	−0.764	3.445
Playground	0.154	1.068	−0.134	0.929	−0.113	0.739
F-statistic	5.900		5.921		4.778	
P-F statistic	0.001		0.001		0.002	
R^2^	0.596		0.597		0.544	
AICc	82.427		82.359		86.156	
RSS	12.525		12.498		14.126	
Durbin-Watson	2.155		2.327		2.294	

**Table 4 ijerph-20-03836-t004:** Overall results of the three types of regression models (MGWR).

MGWR	Model 1: EGR		Model 2: PR		Model 3: FR	
	$\bar{\beta }$	|*t*|	$\bar{\beta }$	|*t*|	$\bar{\beta }$	**|*t*|**
Intercept	−0.042	0.336	0.005	0.036	0.074	0.609
Frost	−0.110	0.621	0.043	0.241	0.075	0.442
Non-farm output	0.542	3.263	−0.339	1.946	−0.530	3.091
Road density	0.014	0.067	0.112	0.511	−0.119	0.577
Altitude	−0.072	0.376	0.641	3.321	−0.552	3.024
Rain	0.503	2.401	−0.042	0.207	−0.724	3.748
Playground	0.146	1.048	−0.122	0.865	−0.126	0.931
tr(*S*)	7.664		7.607		8.754	
Degree of freedom	23.336		23.393		22.246	
R^2^	0.636		0.626		0.690	
AICc	81.813		82.394		81.548	
RSS	11.281		11.581		9.614	

**Table 5 ijerph-20-03836-t005:** Properties of the various influence regions.

Variables	Region (I)	Region (II)	Region (III)
Frost	31.90	12.76	48.19
Non-farm output	95.24	90.55	88.09
Road density	1.66	0.34	0.86
Altitude	1.31	3.13	2.19
Rain	3.02	2.62	2.98
Playground	7.14	6.71	7.15
Number of Provinces	11	7	13

## Data Availability

Not applicable.

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
