# Peer review of "Spatial Heterogeneity and Regional Clustering of Factors Influencing Chinese Adolescents’ Physical Fitness"

_ijerph, 2023, doi:10.3390/ijerph20053836_

Round 1

Reviewer 1 Report

The topic is interesting, however the method of analysis is not described in an easy to comprehend way, and could benefit from simplicity.

Also, I have included few comments within the manuscript PDF attachment.

Author Response

Thank you for considering our manuscript, and your comments concerning our manuscript entitled “Spatial Heterogeneity and Regional Clustering of Factors Influencing Chinese Adolescents' Physical Fitness”. The comments are all valuable and very helpful for improving our paper. We have now carefully reviewed and addressed all the comments, with revisions to the revised manuscript using the "Track Changes" function in Microsoft Word. Details of the changes can be found in the attachment. An abstract of the main revised content is shown below:

  • We have added the reasons for the decline in adolescent physical fitness based on your comments.
  • We have added reasons why the physical fitness of Asian adolescents in particular needs attention based on your comments.
  • We have provided explanations and examples of living environment factors based on your comments.
  • We have provided an explanation and examples of the multidimensional properties of environmental factors based on your comments.
  • We have provided an explanation and examples of the controversial health effects of urbanization based on your comments.
  • We have added references to the formula for the physical fitness test based on based on your comments.
  • We have added clarifications on the meaning of the indicators in the tables and formulas based on your comments.
  • We have explained the influencing factors in the MGWR model and added references based on your comments.
  • We have provided additional clarification on bandwidth, acronyms based on your comments.
  • We have rephrased the semantically unclear sentences in the article based on your comments.
  • We have added reasons for the effect of altitude on physical activity in adolescents based on your comments.
  • We have marked special numbers in the article, table, and figure in brackets based on your comments.
  • We have provided additional clarification on the location of the China Plain based on your comments.

Thank you for everything you have done for us, and we wish you all the best.

We look forward to hearing from you at your earliest convenience.

Reviewer 2 Report

I propose the clear restructuring of the work!

Dividing the paper into clear sections:

-Literature review (There are not enough previous studies are selected that reinforce the need to resume discussion in this study. Much of the information presented in the Results subsection can be used in the Literature Review subsection.

-Hypotheses.

-Clear objectives.

-Methodology.

-Data collection and analysis process.

-Results and discussion.

-Conclusions and planning implications.                                                                                                                                             

Author Response

Thank you for considering our manuscript, and your comments concerning our manuscript entitled “Spatial Heterogeneity and Regional Clustering of Factors Influencing Chinese Adolescents' Physical Fitness”. The comments are all valuable and very helpful for improving our paper. We have now carefully reviewed and addressed all the comments, with revisions to the revised manuscript using the "Track Changes" function in Microsoft Word. Details of the changes can be found in the attachment. An abstract of the main revised content is shown below:

  • We have restructured the article based on your comments.
  • We have added more content and references to the chapter literature review based on your comments.
  • We have transferred some information from the chapter results to the chapter literature review based on your comments.

Thank you for everything you have done for us, and we wish you all the best.

We look forward to hearing from you at your earliest convenience.
